# Addressing Manufacturability and Processability in Polymer Gel Electrolytes for Li/Na Batteries

**DOI:** 10.3390/polym13132093

**Published:** 2021-06-24

**Authors:** Víctor Gregorio, Nuria García, Pilar Tiemblo

**Affiliations:** Instituto de Ciencia y Tecnología de Polímeros, ICTP-CSIC, Juan de la Cierva 3, 28006 Madrid, Spain; ngarcia@ictp.csic.es (N.G.); ptiemblo@ictp.csic.es (P.T.)

**Keywords:** polymer gel electrolytes, composite electrolytes, industrial scaling, lithium batteries, sodium batteries

## Abstract

Gel electrolytes are prepared with Ultra High Molecular Weight (UHMW) polyethylene oxide (PEO) in a concentration ranging from 5 to 30 wt.% and Li- and Na-doped 1-butyl-1-methylpyrrolidinium bis(trifluoromethylsulfonyl)imide (PYR14-TFSI) by a simple procedure consisting of dissolving PEO by melting it directly in the liquid electrolyte while stirring the blend. This procedure is fast, reproducible and needs no auxiliary solvents, which makes it sustainable and potentially easy to scale up for mass production. The viability of the up-scaling by extrusion has been studied. Extrusion has been chosen because it is a processing method commonly employed in the plastics industry. The structure and morphology of the gel electrolytes prepared by both methods have been studied by DSC and FTIR, showing small differences among the two methods. Composite gels incorporation high concentrations of surface modified sepiolite fibers have been successfully prepared by extrusion. The rheological behavior and ionic conductivity of the gels have been characterized, and very similar performance of the extruded and manually mixed gels is detected. Ionic conductivity of all the gels, including the composites, are at or over 0.4 mS cm^−1^ at 25 °C, being at the same time thermoreversible and self-healing gels, tough, sticky, transparent and stretchable. This combination of properties, together with the viability of their industrial up-scaling, makes these gel electrolyte families very attractive for their application in energy storage devices.

## 1. Introduction

Solid electrolytes can make batteries much safer, not only because they prevent liquid electrolytes leaks which are potentially toxic or corrosive, but mainly because they mitigate or avoid dendrite growth and subsequent short-circuits. Dendrite growth occurs not only in Li batteries [1], but also in Na [2], Al [3], Ca [4] and Zn [5] ones (only apparently not in Mg [6]), and thus its mitigation is of interest for a broad battery community. Debate as to the mechanism which impedes the growth of metallic dendrites has taken place in the past. Some authors suggested that dendrite mitigation in solid or quasi solid electrolytes is a mechanical mechanism in which the elastic modulus of the electrolyte plays a role [7,8], and as a matter of fact, in recent times, many strategies to avoid dendrites were largely based on this mechanism [9]. If elastic modulus is the key factor, inorganic solid electrolytes perform well against dendritic growth for they constitute a good mechanical barrier against dendritic growth, but they present poor solid—solid interfaces with the electrodes and are frequently mechanically fragile. Polymer-based electrolytes can be made to wet the electrodes excellently and are mechanically tough, but in homogeneous polymer-based electrolytes where ion diffusion occurs via a Brownian fluid motion, the higher the elastic modulus (or viscosity) of the polymer electrolyte, the lower the ion mobility and so electrochemical performance will worsen. 

Because of this trade-off between elastic modulus (and dendrite suppression) and ion mobility, polymer electrolytes have not been considered as ideal solid-like electrolytes unless accompanied by other components able to stop dendrites, like inorganic particles or phase-separated morphologies [10]. However, there is evidence that even soft polymer-based electrolytes are able to suppress dendritic growth not only in Li batteries [1,11], but also in Al [3] ones. In recent work [12,13] it has been shown that in gel electrolytes prepared with low concentration of UHMW polymers, electroconvection during charging is strongly reduced and hydrodynamic stability much increased, which leads to homogeneous metallic deposits at the anodes and the elimination of dendritic growth. The hypothesis is that electroconvection suppression is uncoupled from ion mobility if the polymer network elastic character predominates enough over the viscous one, and these gels with no dendritic growth still will display liquid-like ionic mobilities. 

In this scenario, polymer gels or highly plasticized polymer electrolytes, which are soft materials with comparatively high ion mobility, can be real candidates for future commercial batteries given that they eliminate dendritic growth and display sufficient electroactivity. Then, as the commercialization of these quasi-solid state batteries (QSSB) gets closer, addressing the sustainable processing and manufacturability at a large scale of solid electrolytes becomes urgent [14]. Over a decade ago, our laboratory initiated a research line on the development of polymer-based solid electrolytes for Li batteries by making use of the processing procedures most common in the polymer industry: extrusion, injection, melt mixing, hot press molding, etc. These methods have many interesting features: first, their scaling for mass production is straightforward; second, they are very fast and employ comparatively low temperatures and no auxiliary solvents; and finally, the resulting electrolytes will be recyclable and reprocessable, since for a material to be processed by melt compounding, it needs to be thermoplastic/thermoreversible. It has been shown that it is possible to design highly plasticized polymer-based electrolytes which can be processed by melt compounding or similar processing methods by using strategies which are well-known in polymer science, like the use of ultra-high molecular weight (UHMW) polymers in combination with physical crosslinkers [15,16,17] or the preparation of polymer blends combining crystalline and amorphous domains [18]. In all cases, together with the polymer, a liquid electrolyte is used, and the ratio between both can be tuned to attain tailored combinations of rheology and ion mobility.

In our previous work [15,16,17,18], the polymer electrolyte blends were considered “highly plasticized” polymer electrolytes in the sense that the polymer was not required to be soluble or thermodynamically miscible in the liquid phase, but simply compatible. Compatibility is a concept much used in polymer science and technology which implies that because of kinetic reasons, a polymer blend will not phase-separate in long time periods. This does not mean that the components are thermodynamically miscible, as polymer miscibility (thermodynamic miscibility) is a rare phenomenon. In this connection, one of the purposes of the ad-hoc modified sepiolite fibers which we have used profusely in our work is acting as a Pickering emulsifier of the liquid electrolyte (cyclic carbonates or ionic liquids-based) which is blended with the polymer to enhance ionic conductivity. These sepiolite fibers stabilize microscopic liquid droplets in the polymer matrix and impede phase separation [18,19] in addition to acting as a mechanical reinforcement and physical crosslinker. As a matter of fact, it is very likely that none of the polymer electrolytes prepared by melt compounding are homogeneous at a molecular scale as their morphology is rather that of a microphase-separated system [19]. When working with compatible polymer blends, the material’s key features are relaxations and phase transitions, whether the polymer is semicrystalline and retains a certain crystalline or on the contrary is fully amorphous, and the temperatures of those relaxations and transitions of the blends (T_g_, T_m_). These T will govern the temperature at which processing can be done and also the mobility of polymer chains at room temperature and hence the likeability of phase separation. When it comes to polymer gels prepared directly by dissolving a polymer in a liquid, the concept is completely different, since in this case thermodynamic miscibility of the polymer in the liquid is required, and not mere compatibility. Not only that, but the rules that apply to understand and tailor the material prepared are not those empirical rules of polymer processing, but rather the laws of polymers in solution first introduced by Flory [20]. The key parameters to tune the properties of polymer gels are the solubility of the different components and polymer molecular weight and concentration. Those last ones will govern the threshold for polymer entanglements and gel formation. Thus, while it is possible to prepare highly plasticized polymer electrolytes, even with very large (<60 wt.%) fractions of liquid phase by using a combination of non-miscible polymer/liquid, this is not possible in a polymer gel. 

Gel polymer electrolytes for Li, Na, or Zn are not usually prepared by melt compounding or similar solvent-free procedures but by casting from solvents, and the gels are most often chemical, i.e., a chemically bonded polymer network is created which does not soften by increasing temperature [21]. However, by choosing pairs of soluble polymer/liquid electrolytes, it is possible to prepare these gels by direct dissolution. Very recently we underwent the preparation of the chloroaluminate polymer gels for Al secondary batteries. Chloroaluminates are very sensitive to ambient humidity, producing HCl if exposed, and they must be handled inside a glovebox. Thus, melt compounding in an extruder or similar open air preparation procedures were out of question. Though polymer solubility in chloroaluminates is not straightforward [22], we found that several polymers, including poly(ethylene oxide) (PEO) can be used to prepare chloroaluminate gels by a simple procedure which basically consists of stirring manually the liquid electrolyte and the polymer while a slow temperature ramp up to the melting point of the polymer is applied [23]. This procedure requires no auxiliary solvents nor intermediates, produces thermoreversible gels if UHMW polymer is employed, and is feasible by stirring as long as the polymer concentration is low. 

Inspired by this, we decided to study the viability of extrusion for the upscaling of gel electrolytes which can be prepared by simply melting PEO while mixing it in Li and Na electrolytes. While in previous work [15,16] we explored melt compounding in the polymer concentration range from 50 down to 30 wt.% of compatible polymer electrolytes, in this work we explore the use of scalable processing procedures in the polymer concentration range from 5 to 30 wt.% for the solvent-free preparation of polymer gel electrolytes. 

## 2. Materials and Methods

### 2.1. Materials

Li and Na bis(trifluoromethylsulfonyl)imide salts (LiTFSI and NaTFSI, respectively) were supplied by Sigma-Aldrich (St. Louis, MO, USA) and dried in vacuum before used. 1-Butyl-1-methylpyrrolidinium bis(trifluoromethylsulfonyl)imide (PYR14-TFSI, Solvionic, Toulouse, France) and PEO (Mw = 5 × 10^6^ g mol^−1^) supplied by Sigma-Aldrich were used as received. Sepiolite was kindly supplied by TOLSA S.A. (Madrid, Spain) and modified with D-α-tocopherol polyethylene glycol 1000 succinate (TPGS) from Sigma Aldrich. The surface modification procedure is described in previous works [24]. In what follows, surface modified sepiolite is named TPGS-S.

### 2.2. Preparation of the Gel Electrolytes

First, the Li or Na salt is well dissolved in PYR14-TFSI to obtain the liquid electrolyte Li(PYR14-TFSI) or Na(PYR14-TFSI). After that, the dissolution is mixed with PEO following two different methodologies:

Manual mixing: This methodology previously reported in our group [23] consists of mixing the PEO with the liquid electrolyte at room temperature and stirring it non-stop while increasing the T to 70 °C (just above the melting temperature of the PEO). The temperature ramp takes about 10 min, and stirring at 70 °C continues for another 10 min.

Extrusion processing: PEO powder is well dispersed by hand in the liquid electrolyte, after which the mixture is melt compounded in an extruder Haake minilab (Thermo Fisher Scientific, Waltham, MA, USA) at 75 °C and 80 rpm. The residence time of the mixture in the extruder is 8 min. TPGS-S is dispersed in the liquid electrolyte by stirring for 1 h at room temperature before adding the PEO.

A schematic representation of the materials employed and the processing procedure is presented on Scheme 1.

In this way, the electrolytes in Table 1 were prepared. The nomenclature employed in this work is PEOx(m/e) where x is the polymer wt.% in the mixture and the letter m for manual mixing or e for the extruded samples. If nothing is added, the electrolyte contains LiTFSI. When the mixture is made with NaTFSI, Na is included in the sample name. Electrolytes with the filler TPGS-S were prepared only by extrusion and are named PEO30e/S10 (10 wt.% TPGS-S) and PEO30e/S20 (20 wt.% TPGS-S).

### 2.3. Characterization

IR spectra of the electrolytes was recorded with a FTIR Perkin-Elmer Spectrum-Two (PerkinElmer, Waltham, MA, USA), measuring 4 scans with a resolution of 4 cm^−1^.

Differential Scanning calorimetry curves have been obtained with a TA Instruments Q100 (TA Instruments, New Castle, DE, USA). The sample was cooled down from RT to −80 °C as quickly as possible to quench the polymer structure. Then the sample was heated to 120 °C, follow by a cooling cycle to −80 °C and a new heating cycle to 120 °C. All cycles were done with a heating—cooling speed of 10 °C min^−1^.

Rheological measurements were performed in an AR-G2 rheometer with a 25 mm diameter stainless steel geometry in a frequency range of 0.01–100 rad s^−1^. Measurements were made at 25 and 75 °C. Reported electrolyte mechanical properties were estimated with the curves at 75 °C in order to eliminate the polymer crystallinity effect. 

Self-creep experiments were done as follows: electrolytes’ films were sandwiched between two gold electrodes of 20 mm of diameter and placed on a heating plate with 0.5 kg on top at 70 °C and 90 °C for 20 min at each temperature.

The ionic conductivity of the electrolytes was measured in a NOVOCONTROL Concept 40 broadband dielectric spectrometer (Novocontrol Technologies GmbH, Montabaur, Germany). The measurements were carried out in a frequency range from 0.1 to 10^7^ Hz and a temperature window on heating from −50 to 90 °C every 10 °C and on cooling from 85 to 25 °C every 10 °C. The electrolytes were placed between two stainless steel electrodes with a Teflon scaffold to avoid creep during measurement. 

For the self-healing experiments, the samples were hot pressed at 80 °C to obtain a thin film of 400 μm. Then, the film was cut from top to bottom and its healing was tracked taking images by photography and with an optical profilometer Zeta Instrument model Z-20 (KLA, San Diego, CA, USA).

## 3. Results and Discussion

Li and Na electrolytes have been successfully prepared by extrusion and by manual stirring. Note that to prepare gels by manual mixing or melt compounding without the addition of other auxiliary solvents, the first step is the dissolution of the Li or Na salt in the chosen liquid, in this case PYR14-TFSI. Subsequently the polymer is dissolved in that liquid electrolyte. When, instead of gels, highly plasticized PEO electrolytes are prepared by extrusion, the first step is the mixture of the Li salt with PEO to ensure an intimate blending between the polymer and the salt which avoids crystallization of PEO, especially when the anion TFSI is present. 

Scheme 2 collects melt-compounded polymer-based electrolytes published throughout the last few years by our group, in comparison with those studied in this work. As mentioned in the Introduction, there is a conceptual difference among both materials in that polymer blending does not imply that the polymer is soluble in the liquid electrolyte, only that it is compatible, and careful design and the use of compatibilizers will allow for these blends to be stable for long periods of time. The morphology of these blends is that of a microscopic phase separation. Gels prepared without auxiliary solvents, however, require the solubility of the polymer in the liquid electrolytes and the material will be much more homogeneous at a molecular level. As will be shown along this work, the different concept produces also different properties even among electrolytes with very similar formulation, as those close to 30 wt.% PEO in Scheme 2. 

Even if the wt.% of PEO is low in some of these gels, the combination of the chain entanglements produced when using these UHMW PEO and the well-known crosslinking effect of Li and Na cations in the presence of this polymer, allows to obtain gels which are stretchable, thermoreversible/thermoplastic and self-healing, and also sticky and transparent, as shown in Figure 1. The gel electrolytes with higher PEO loading like the PEO30 series with or without TPGS-S are self-standing and flexible electrolytes with the mechanical properties of a solid (as shown in Figure 1a), but with the ability to wet the electrodes of a viscous liquid. Their self-healing ability is particularly interesting and it appears illustrated in Figure 1b,c. Figure 1b shows pictures of a PEO30e slab, which has been cut. Throughout the next 40 min the cut heals progressively by a combination of creep and physical crosslinking. Creep alone is not responsible for the healing, as the slab retains its dimensional stability along the same time period, showing no remarkable creep at its edges. Figure 1c shows the same cut at a magnification of 5×, recorded by optical profilometry. In the Appendix A, a GIF built with the profilometry images is included, in which images are recorded every 2 min and comprise the whole process, which lasts 40 min. Figure 1d shows the stretchability of the different gels. The properties illustrated in Figure 1 are shared by all the electrolytes, the only difference being the increase in elastic modulus of gels as either the polymer PEO or the filler TPGS-S is added at higher concentrations. Their toughness, self-healing ability, and flexibility make them really easy to handle and very robust, something very attractive for really applicable materials. 

DSC and FTIR have been used to comparatively study some selected electrolytes prepared by stirring and by extrusion, and the results appear in Appendix A, respectively. The DSC show that PEO has lost its crystallinity in all the electrolytes; however, in the electrolytes prepared with 5 wt.% of PEO, a phase transition is seen from −30 to 20 °C in PEO5m and PEO5e and in the region −60 to −20 °C in the PEO5m/Na and PEO5e/Na gels. This reveals that in these gels, a phase separation exists. This phase separation is less conspicuous in the extruded electrolytes, evidencing that extrusion produces better blends. Interestingly, a 15 wt.% of PEO is enough to eliminate completely those liquid phase transitions.

The FTIR of the region 750–800 cm^−1^, where υ(S-N) of the TFSI anion appears is very useful to discriminate between free ions, ion pairs and salt aggregates, for the stronger the interaction between both TFSI and Li, the higher the wavenumber. The FTIR of all electrolytes in this work appears in Appendix A, showing a very similar distribution of free ions and ion pairs in them. In Appendix A, the 30 wt.% gels display bands in the 960 and in the 1100 cm^−1^ regions belonging to the methylene angle deformation and the backbone stretching modes of PEO, respectively. In those electrolytes with PEO concentration 15 wt.% or under, the backbone vibrations of PEO are too weak to be detected. The preparation procedure, manual or extrusion, does not produce visible differences in the FTIR in electrolytes prepared with the same PEO concentration.

### 3.1. Rheology

Figure 2 show the rheological curves at 75 °C of all the electrolytes prepared in the range of frequencies from 0.01 to 100 rad s^-1^ Figure 2a shows the storage moduli (G′) and loss moduli (G″) of Li gel electrolytes. The increase in PEO concentration from 5 to 30 wt.% increases the moduli substantially as can be expected. The effect of TPGS-S concentration on shear moduli is seen in the PEO30 series in Figure 2b. The filler acts as mechanical reinforcement, increasing the elastic modulus, similarly to the behavior reported in previous works on PEO electrolytes prepared with higher polymer concentration [25]. Figure 2c shows G′ and G″ of the Na gel electrolytes, which are slightly under those of analogous Li gel electrolytes in Figure 2a. The lower storage moduli in comparison with the Li gels is very likely produced by the lower concentration of metallic cations present in the electrolyte, which implies less crosslinking points between polymer chains. Interestingly, in a first approach, in all the gels of Figure 2, very similar rheological curves are found for the gel electrolytes prepared manually and by extrusion which suggests that PEO is well dissolved by using both procedures.

For a quantitative comparison among the gels rheology, three parameters have been chosen: the angular frequency at which the elastic and loss moduli become the same, i.e., under which the material behaves as a viscoelastic liquid ω_G__′=G″_, called the crossover frequency; the elastic modulus G′ at 100 and 10 rad s^−1^, which describes the stiffness of the material at high and low frequencies; and the ratio $\frac{G^{\prime }}{G^{\prime \prime }}$ calculated at 0.05 rad s^−1^. These results appear in Table 2.

The ω_G′=G″_ of the gels prepared with 5 wt.% of PEO does depend slightly on the preparation procedure. The extruded Li or Na gels with 5 wt.% of PEO (PEO5e and PEO5e/Na) have ω_G′=G″_ at 0.01 rad s^−1^, while the manually mixed ones (PEO5m and PEO5m/Na) are at ω < 0.01 rad s^−1^. The ω_G′=G″_ depends on the number of entanglements which in turn depends on the polymer Mw. This slight difference found between extruded and manual gels with 5 wt.% may be caused by a certain polymer chain breaking during extrusion that is known to occur when processing polymers in this way as a consequence of mechanical chain scission under shear. All gels prepared with higher PEO concentration have ω_G′=G″_ < 0.01 rad s^−1^, and it is not possible to detect crossover frequency differences in them.

Figure 3a represents G′ at 100 rad s^−1^ and the ratio $\frac{G^{\prime }}{G^{\prime \prime }}$ at 0.05 rad s^−1^ as a function of the PEO plus TPGS-S concentration in wt.%. G′ increases as the content of Li(PYR14-TFSI) decreases, i.e., as the wt.% of PEO increases first from 5 to 30 wt.%, and then as the wt.% of TPGS-S increases from 10 to 20 wt.%. TPGS-S is then acting as a mechanical reinforcement in the PEO30e/S10 and PEO30e/S20 gels. The preparation procedure has no detectable influence on these gels′ G′, and in fact, reproducibility of this parameter in the different gels is excellent. The presumed slight degree of polymer chain scission caused by melt compounding has no effect on G′, which is not surprising as this rheological parameter depends more on the concentration of PEO than on its molecular weight. 

As regards the ratio $\frac{G^{\prime }}{G^{\prime \prime }}$, Figure 3a reflects how it increases with the PEO concentration. This is expectable since this ratio estimates the predominance of the elastic over the viscous character, which depends on the PEO concentration. Opposite to G′, the ratio $\frac{G^{\prime }}{G^{\prime \prime }}$, at 0.05 rad s^−1^ does depend clearly on the preparation procedure and is always higher in manually mixed samples, i.e., extruded gel electrolytes (violet open circles in Figure 3) are slightly more viscous in character than manually mixed ones (orange open circles) of the same composition. Again, this difference can be caused by a slight chain breaking during extrusion, which moves the crossover to a higher frequency.

Surprisingly, this ratio initially increases on adding TPGS-S (PEO30e/S10), but for higher TPGS-S concentration it decreases. This suggests that the filler is acting as a mechanical reinforcement but it is not promoting crosslinking as compared to the PEO30e electrolyte. In this connection, Figure 3b shows images of a self-creep experiment performed with PEO15e, PEO30e, and PEO30e/S10. The PEO30e gel is able to withstand a 0.5 kg load for 20 min and at 70 °C without flowing, demonstrating its solid-like character. Interestingly, highly plasticized electrolytes prepared by melt compounding with higher wt.% of PEO of this same molecular weight as PEO30e need the modified filler TPGS-S to display solid-like character. This was shown in electrolytes prepared with EMIFSI [16] or with ethylene carbonate [19] which could not withstand the self-creep experiment and suffered flow. This occurs because in these blends, the components are compatible but there is no true solubility of PEO in the liquid electrolyte. Hence, these blends require TPGS-S for compatibilizing the liquid and solid phases and to crosslink the PEO. The PEO15e gel does flow when subjected to the same experiment, for the concentration of PEO is too low.

The gels prepared in this work do not need TPGS-S as compatibilizer because the polymer and the liquid are miscible and no phase separation will occur. TPGS-S is superfluous as physical crosslinker in gels with 30 wt.% of PEO because the gel is physically crosslinked by the polymer entanglement and the interaction of the Li cation with the PEO. Thus, the polymer composite electrolytes which are highly loaded with TPGS-S (PEO30e/S10 and PEO30e/S20) are mechanically reinforced (G′ increases) but they do not display longer creep times than PEO30e, as shown in Figure 3a. Though TPGS-S is not required for compatibilizing or crosslinking, its stiffening effect may be very desirable because in former work [25] it was shown that TPGS-S concentrations in the range of 10–20 wt.% increased capacity retention in solid-state coin cells Li-LiFePO. 

The proposed effect of polymer chain breaking on melt compounding is not very strong, and once detected, can be controlled by optimizing formulation and melt-compounding conditions. In fact, this chain breaking does not produce further radical degradation of the polymer in the electrolytes which contain TPGS-S, for TPGS is a well-known radical stabilizer of PEO chain degradation, employed in the pharmaceutical industry for its melt-compounding. This allows proposing extrusion as valid for the upscaling of gel electrolyte preparation using UHMW polymers (the most sensitive to chain scission under shear) as with those in this work. In fact, the rheology presented in this work suggests that in gels where very strong liquid/polymer interaction occurs, but where the polymer dissolution does not occur until the polymer melts, the key step for gel formation is the homogeneous dispersion at room temperature of the polymer powder in the liquid. Any shear/temperature processing method can be subsequently used to prepare and process the gels.

### 3.2. Ionic Conductivity

Table 2 collects the values of σ at 25 and 75 °C for all electrolytes. Figure 4 shows how σ decreases as the liquid electrolyte content decreases in the gel, as can be expected. Extruded gels in all the PEO concentration range have σ values only slightly lower than manually prepared gels of analogous composition. The reduction of σ is gentle with PEO concentration, and for example, PEO30e is a gel electrolyte which combines an interesting σ of 0.6 mS cm^−1^ at 25 °C with dimensional stability and self-healing ability, as shown in Figure 1c, and at the same time can be prepared by extrusion. These σ are in the range of those published for compatible blends with similar formulation such as PMPTFSI−4 in reference [16], which is 0.8 mS cm^-1^ Thus, gels in comparison with compatible blends seem to display similar σ, which depends on the liquid electrolyte concentration, and present higher mechanical stability with more reproducible mechanical properties than in the case of highly plasticized electrolytes, where the properties depend on the quality of the mixture, as is studied in EMIFSI samples in ref [16].

The addition of TPGS-S to PEO30e produces a slight and progressive σ decrease caused by the liquid electrolyte content decrease and the tortuosity introduced by the inorganic fiber. Nonetheless, the composite gel electrolytes PEO30e/S10 and PEO30e/S20 also retain sufficient σ for being applicable as gel electrolytes. At the same time, they possess high TPGS-S concentration and significant stiffness, which as mentioned before can be beneficial for improving capacity retention in solid-state coin cells Li-LiFePO_4_ [25].

In Figure 5, σ(T) in the range −50 to 90 °C appears. On-heating scans appear in solid symbols and on cooling in open symbols. All polymer gels are well fitted by a VFT function, i.e., they behave as viscous liquids in all the T range, with no step-like σ variation caused by phase transitions.

The decrease of σ with T is not very strong between 70 °C and 10 °C, about an order of magnitude, being still about 0.1 mS cm^−1^ at 0 °C in most of the gels. Under 0 °C, σ decreases very quickly, being at −40 °C in the range of 10^−4^ mS cm^−1^. On-cooling σ values are very slightly higher than on-heating in almost all the gels, which means that there is a non-instantaneous rearrangement of the gel structure. Little difference is seen in the σ(T) of extruded and manually mixed gel electrolytes.

## 4. Conclusions

In this work gel electrolytes are prepared with UHMW PEO and Li- and Na-doped PYR14-TFSI by a simple procedure consisting of melting PEO directly in the liquid electrolyte while stirring the blend, producing the dissolution of the polymer. The gels thus prepared are stretchable, sticky and self-healing. The preparation procedure is fast, reproducible and needs no auxiliary solvents, which makes it sustainable and potentially easy to scale up for mass production. The viability of the up-scaling has been attempted by extrusion, a processing method which is among the most commonly employed in the plastics industry. Gels with PEO concentration in the range 5 to 30 wt.% have been successfully prepared by manual stirring and by extrusion, and their molecular structure, degree of mixing, rheology, and ionic conductivity have been compared.

The FTIR of the gel electrolytes show that as regards the dissolution of the salt (LiTFSI or NaTFSI), there is no difference among manually stirred or extruded samples, and in all cases the salt dissolution is very good. The DSC scans reveal that in the electrolytes prepared with the lowest PEO concentration (5 wt.%), a certain phase separation exists which is less important in the melt-compounded electrolytes. The ionic conductivity of the extruded and manually mixed gels is very similar, and though it depends on PEO concentration, this dependence is gentle in the range of 5 to 30 wt.%. Even the addition of TPGS-S in high concentrations (up to 20 wt.%) does not bring ionic conductivity dramatically down.

The rheology curves show that in those electrolytes prepared with the lowest PEO concentration (5 wt.%), the crossover frequency is seen at very low frequencies (ω_G′=G″_ at 0.01 rad s^−1^) in the extruded gels PEO5e and PEO5e/Na, while it is not seen in the manually mixed ones (PEO5m and PEO5m/Na) because it is at ω < 0.01 rad s^−1^. This is attributed to a certain chain scission of the UHMW PEO occurring during the melt compounding. This chain scission does not impact significantly the shear moduli of any of the gels, and only slightly the balance between elastic and viscous shear moduli. Thus, its effect, though detectable, is gentle, and can be controlled by optimizing formulation and melt compounding conditions. On the other hand, DSC proves that extrusion produces better blending than manual mixing.

The results presented in this work allow for the proposal of extrusion as a valid processing method for the up-scaling of gel electrolytes as with those in this work.

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
