# Peer review of "Addressing Manufacturability and Processability in Polymer Gel Electrolytes for Li/Na Batteries"

_polymers, 2021, doi:10.3390/polym13132093_

Round 1
Reviewer 1 Report
In this paper authors present the work entitled "Addressing manufacturability and processability in polymer electrolytes for Li/Na batteries. From gels to highly plasticized polymer electrolytes". The study, understanding and then the further technological evaluation of gel polymers for energy applications is definitely relevant.
In my opinion the work is OK, well conducted with sound results, thus I would like to recommend publication as is.
Author Response
Thank you for your assessment.
Reviewer 2 Report
In this work, the authors reported manufacturability and processability in polymer electrolytes for Li/Na batteries. The viability of the up-scaling by extrusion has been studied. The objectives of the paper have been stated in the introduction section, and the methods and results are presented in some way. However, the research methodology and content in this manuscript seems similar with previous work [15,22]. The manuscript needs revision considering the following aspects:
The difference between this work and the previous ones [15,22] needs to be elaborated more. What is the novelty of this work except the different polymer concentration range?
The title ‘batteries. From’ is confusing.
In the introduction, it is worth mentioning that the elastic modulus actually changes during the lithiation and delithiation process, i.e., the changes of concentration of Li-ion. It would be helpful to add: The effects of elastic stiffening on the evolution of the stress field within a spherical electrode particle of lithium-ion batteries; Effects of composition-dependent modulus, finite concentration and boundary constraint on Li-ion diffusion and stresses in a bilayer Cu-coated Si nano-anode.
In the introduction, the extrusion technique involving compressive plastic deformation needs to be briefly introduced and referenced [International Journal of Machine Tools and Manufacture 126 (2018) 27-43], which is a commonly used materials processing technology with full potential for up-scaling.
The y-axis in Fig. 5 needs to be properly adjusted to better shown the values of data points.
Different ‘symbols + line types should be used to better differentiate different types of data points. Currently they are only differentiated by different colours.
Reviewer 3 Report
- Very small change on the Ionic conductivity in Figure 5 for different samples. How to explain this? More discussion is required for this part.
- How about the structural stability towards organic solvents that are often used in batteries?
- To highlight the manufacturability and processability, is there any consideration on production costs?
